# Assessment of a Large-Scale Unbiased Malignant Pleural Effusion Proteomics Study of a Real-Life Cohort

**DOI:** 10.3390/cancers14184366

**Published:** 2022-09-08

**Authors:** Sara Zahedi, Ana Sofia Carvalho, Mostafa Ejtehadifar, Hans C. Beck, Nádia Rei, Ana Luis, Paula Borralho, António Bugalho, Rune Matthiesen

**Affiliations:** 1iNOVA4Health, NOVA Medical School (NMS), Faculdade de Ciências Médicas (FCM), Universidade Nova de Lisboa, 1150-082 Lisbon, Portugal; 2Department of Clinical Biochemistry, Odense University Hospital, 5000 Odense, Denmark; 3Hospital CUF Descobertas, CUF Oncologia, 1998-018 Lisbon, Portugal

**Keywords:** biomarker, diagnosis, malignant, lung cancer, proteomics, risk models, pleural effusion

## Abstract

**Simple Summary:**

Pleural effusion (PE) occurs as a consequence of various pathologies. Malignant effusion due to lung cancer is one of the most frequent causes. A method for accurate differentiation of malignant from benign PE is an unmet clinical need. Proteomics profiling of PE has shown promising results. However, mass spectrometry (MS) analysis typically involves the tedious elimination of abundant proteins before analysis, and clinical annotation of proteomics profiled cohorts is limited. This study compares the proteomes of malignant PE and nonmalignant PE, identifies lung cancer malignant markers in agreement with other studies, and identifies markers strongly associated with patient survival.

**Abstract:**

Background: Pleural effusion (PE) is common in advanced-stage lung cancer patients and is related to poor prognosis. Identification of cancer cells is the standard method for the diagnosis of a malignant PE (MPE). However, it only has moderate sensitivity. Thus, more sensitive diagnostic tools are urgently needed. Methods: The present study aimed to discover potential protein targets to distinguish malignant pleural effusion (MPE) from other non-malignant pathologies. We have collected PE from 97 patients to explore PE proteomes by applying state-of-the-art liquid chromatography-mass spectrometry (LC-MS) to identify potential biomarkers that correlate with immunohistochemistry assessment of tumor biopsy or with survival data. Functional analyses were performed to elucidate functional differences in PE proteins in malignant and benign samples. Results were integrated into a clinical risk prediction model to identify likely malignant cases. Sensitivity, specificity, and negative predictive value were calculated. Results: In total, 1689 individual proteins were identified by MS-based proteomics analysis of the 97 PE samples, of which 35 were diagnosed as malignant. A comparison between MPE and benign PE (BPE) identified 58 differential regulated proteins after correction of the *p*-values for multiple testing. Furthermore, functional analysis revealed an up-regulation of matrix intermediate filaments and cellular movement-related proteins. Additionally, gene ontology analysis identified the involvement of metabolic pathways such as glycolysis/gluconeogenesis, pyruvate metabolism and cysteine and methionine metabolism. Conclusion: This study demonstrated a partial least squares regression model with an area under the curve of 98 and an accuracy of 0.92 when evaluated on the holdout test data set. Furthermore, highly significant survival markers were identified (e.g., PSME1 with a log-rank of 1.68 × 10^−6^).

## 1. Introduction

Pleural effusion (PE) is an abnormal accumulation of fluid in the pleural cavity. According to Light’s criteria, PE can be clinically classified as either exudate, which is most frequently caused by lung pleura and systemic disorders, or transudate, which is typically caused by cardiac, kidney, or liver failure [1,2]. Compared to a tissue biopsy, PE is obtained through a less invasive and uncomfortable process [3]. Malignant pleural effusion (MPE) reflects the dissemination of cancer cells to the pleura (pleural metastasis), disrupting normal fluid turnover [4].

Clinical discrimination of PE types is crucial and the gold standard method for MPE detection is cytology of the PE or histology of the pleural membrane by closed biopsy or by thoracoscopy, the latter being more invasive. Although pleural fluid cytology has high specificity, the sensitivity is moderate, 58.2% (95% CI 52.5% to 63.9%) [5]. An imprecise diagnosis makes the discrimination of disease stages more difficult, and it is an obstacle to defining a therapeutic strategy [6].

MPE, a frequent clinical issue in cancer patients, can result from both primary thoracic tumors and a metastatic dissemination in the chest [7]. In about 10% of cases, MPE is the first sign of the disease. Despite improvements in treatment choices, the prognosis is still grim, and the typical survival time after a MPE diagnosis is 4 to 9 months [7]. The clinical management of this condition will be substantially facilitated by biomarkers for survival in patients with MPE [8]. However, current large-scale mass spectrometry-based proteomics studies has not assessed survival of protein biomarkers. Advanced malignancies frequently experience MPE, which lowers quality of life and restricts available treatments.

Lung cancer accounts for 36.0% of MPEs, followed by breast cancer and lymphoma [9]. Around 40% of lung cancer patients with exudative PE are undetected by current thoracentesis and require thoracoscopy for detection [10]. Currently, detection of lung cancer implies late stage and poor prognosis. However, an early detection of lung cancer by an innovative method may lead to a decrease in cancer lethality.

Lung cancer has become the top-ranked leading cause of cancer death globally, accounting for 1.80 million deaths in 2020 (18%) [11]. Two main histological subtypes of lung cancer are small cell lung cancer (SCLC) and non-small-cell lung cancer (NSCLC). NSCLC is usually diagnosed in 85% of lung cancer cases and includes three main subgroups: squamous cell carcinoma (SCC), adenocarcinoma (ADC), large-cell carcinoma (LCC) and some rare subgroups [12]. Despite the advances in understanding lung cancer biology and the implementation of innovative treatment methods such as tyrosine kinase inhibitors and immunotherapies [13], lung cancer has remained a critical clinical issue due to late-stage diagnosis in over 62% of the cases [14]. More importantly, early detection of patients would raise the 5-year overall survival rate from 18% to 55%, giving the patients a chance of being cured by surgery [15]. Vast effort has been invested in attempting to find new methods to diagnose MPE in the early stages. However, the most efficient strategy to overcome the high mortality rate and poor prognosis is to find an efficient panel of biomarkers [16,17,18]. Therefore, there is an urgent demand to seek potential biomarkers to distinguish MPE from benign lung diseases. 

A major limitation of lung tissue biomarkers screening is the sample accessibility, since the procedures are generally highly invasive, particularly for patients in late stages [3]. Another challenging factor is tissue heterogeneity, which adds excessive variability in predicted outcome [19]. To overcome these issues, there has been an increased investment in identifying potential protein biomarkers in body fluids, such as blood, urine, bronchoalveolar lavage and PE [20,21,22,23,24]. The commonly used biomarkers of body fluids in clinical panels such as Carcinoembryonic antigen (CEA), CA-125, CA 15-3 and cytokeratin fragments showed low efficacy in detecting lung cancer [25]. The design of new screening tests, based on more reliable targets, e.g., identified by more advanced and sensitive techniques, is essential.

The proteome profile of lung cancer’s pleural fluid comprises proteins originating from the complex tumor microenvironment, including immune cells, cancer cells, epithelial cells, etc. [26,27]. PE has been investigated by metabolomics and proteomics studies and results are promising in terms of potential biomarkers [28,29,30]. However, more and larger clinical cohorts are needed as well as detailed clinical annotation of samples such as survival data.

Previous clinical cohorts analyzing the proteome of MPE, suggest that a further investigation of PE proteome will provide an overview of lung cancer cell mechanism and its associated signaling cascades [31,32]. Investigations have addressed protein patterns in clinical conditions such as tuberculosis [33]. Furthermore, a number of proteomics studies have been focusing on malignant pleural mesothelioma to determine protein biomarkers by using different sample types [34,35,36]. Other studies analyzed proteome profiling of MPE compared with benign inflammatory diseases [37]. Few studies analyzed patients with suspected MPE with undefined pathogenesis to discover diagnostic markers [38]. However, in this study, we collected PE samples through various etiologies to obtain a well annotated real-life clinical cohort with the aim of identifying biomarkers for MPE and survival.

The present study reveals original findings concerning functional characterization of PE across pathologies, based on PE proteomics profiling, and identifies protein markers strongly correlated with survival. Classification performance of comparative models based on PE proteomics data was also evaluated. Moreover, our results in terms of predicting poor prognosis are concurrent with previous studies.

## 2. Materials and Methods

### 2.1. Patient Samples

PE samples were obtained from patients with lung cancer, suspected lung cancer, mesothelioma cancer, other types of cancers and non-malignant patients at CUF Descobertas hospital, in Lisbon, from 2019 to 2021. The study protocol was approved by NOVA Medical School Ethics Committee (Registry number nr.85/2018/CEFCM) and by the ethical committee from CUF Descobertas hospital. All patients or relatives signed a written informed consent. The samples used for this study are normally discarded samples, thus there is no added inconvenience for the patients. The project was conducted according to the ethical rules of the Helsinki declaration. All the information regarding human patients was anonymized. Inclusion and exclusion criteria of the patients were clearly defined. Patients under 18 years, mentally disabled and patients too weak to evaluate the informed consent were excluded from the study. All patients, in the collaborating institution, able to sign the informed consent were enrolled. All pleural fluid samples were obtained from patients undergoing a thoracocentesis at CUF Descobertas hospital and collected in sterile 50 mL tubes. The samples were subjected to low spin centrifugation (800 rpm) to remove cells. The cell pellet and supernatant samples were stored at −80 °C until further analysis. A total number of 97 PE samples were collected which were later diagnosed, resulting in the following diagnosis: 35 malignant patients, 5 suspected malignant and 57 non-malignant cancer (benign) patients. MPE was confirmed by clinical positive cytology reports or by further invasive diagnostic tests (e.g., pleural biopsy). Non-malignant patients that were referred in this study as benign were further categorized into exudate and transudate benign groups. Benign patients were diagnosed with diseases such as pneumonia, thromboembolism, entrapped lung, inflammatory and auto-immune diseases, heart failure, and renal failure. On the other hand, the malignant group included patients diagnosed with lung cancer and patients diagnosed with other malignancies such as breast, gynecologic, gastric and colorectal cancer.

### 2.2. Peptide Preparation

The frozen PE specimens were thawed and then centrifuged at 3200× *g*, 10 min, 4 °C to pellet cell debris. Next, PE were precipitated with ice cold acetone and proteins resuspended in solution containing RIPA buffer, 4% SDS. Proteins were reduced with 0.1 M dithiothreitol (DTT) and loaded onto 30K spin columns and washed with 8 M Urea 0.1 M HEPES, pH 8.0. Then proteins were alkylated with 50 mm iodoacetamide. Prior to trypsin digestion overnight at 37 °C, proteins were equilibrated with ammonium bicarbonate buffer.

### 2.3. Protein Measurements

Protein concentrations in PE were measured by using a Bicinchoninic acid (BCA) protein assay kit (Pierce Biotechnology, Rockford, IL, USA) according to the manufacturer’s instructions. Bovine Serum Albumin (BSA) was used as the reference standard to generate the standard curve of the BCA protein assay. Additionally, the protein quantifications were validated by SDS-PAGE Coomassie staining.

### 2.4. Mass Spectrometry Analysis

Samples were analyzed by mass spectrometry-based proteomics using a nano-LC-MSMS (Dionex RSLCnano 3000) coupled to an Exploris 480 Orbitrap mass spectrometer (Thermo Scientific, Hemel Hempstead, UK) as previously described [39]. In brief, samples were loaded onto a custom made fused capillary pre-column (2 cm length, 360 µm OD, 75 µm ID, flowrate 5 µL per minute for 6 min) packed with ReproSil Pur C18 5.0 µm resin (Dr. Maish, Ammerbuch-Entringen, Germany), and separated using a capillary column (25 cm length, 360 µm outer diameter, 75 µm inner diameter) packed with ReproSil Pur C18 1.9-µm resin (Dr. Maish, Ammerbuch-Entringen, Germany) at a flow of 250 nL per minute. A 56 min linear gradient from 89% A (0.1% formic acid) to 32% B (0.1% formic acid in 80% acetonitrile) was applied. Mass spectra were acquired in positive ion mode in a data-dependent manner by switching between one Orbitrap survey MS scan (mass range *m*/*z* 350 to *m*/*z* 1200) followed by the sequential isolation and higher-energy collision dissociation (HCD) fragmentation and Orbitrap detection of fragment ions of the most intense ions with a cycle time of 2 s between each MS scan. MS and MSMS settings: maximum injection times were set to “Auto”, normalized collision energy was 30%, ion selection threshold for MSMS analysis was 10,000 counts, and dynamic exclusion of sequenced ions was set to 30 s.

### 2.5. Database Search

The obtained data from the 214 LC-MS runs of 97 PE samples were searched using VEMS [40] and MaxQuant [41]. Of the 97 PE samples 91 were analyzed as duplicates, two were analyzed four times, and four were analyzed six times. The MSMS spectra were searched against a standard human proteome database from UniProt (3AUP000005640). Permuted protein sequences, where Arg and Lys were not permuted, were included in the database for VEMS. Trypsin cleavage allowing a maximum of 4 missed cleavages was used. Carbamidomethyl cysteine was included as fixed modification. Methionine oxidation, lysine and N-terminal protein acetylation, deamidation of asparagine, serine, threonine and tyrosine phosphorylation, diglycine on lysine, and methylation on lysine were included as variable modifications. Five ppm mass accuracy was specified for precursor ions and 0.01 *m*/*z* for fragment ions. The false discovery rate (FDR) for protein identification was set at 1% for peptide and protein identifications. No restriction was applied for minimal peptide length for VEMS search. Identified proteins were divided into evidence groups as defined by Matthiesen et al. [42].

### 2.6. Statistical Analysis and Machine Learning

Quantitative data from MaxQuant and VEMS were analyzed in the R statistical programming language. Protein LFQ and protein spectral counts from the two programs were preprocessed by three approaches: (1) removing common MS contaminants followed by log_2_(x + 1) transformation, (2) removing common MS contaminants followed by log_2_(x + 1) transformation and quantile normalization, (3) removing common MS contaminants followed by log_2_(x + 1) transformation, quantile normalization and abundance filtering to optimize overall Gaussian distribution of the quantitative values. Protein LFQ values were subjected to statistical analysis utilizing R package limma [43] where contrast between MPE versus BPE and MPE versus EBPE were specified (Appendix A).

Correction for multiple testing was applied using the method of Benjamini and Hochberg [44]. Volcano plot was plotted with ggplot followed by annotating data points with potential contaminant proteins from erythrocyte, platelet and coagulation extracted from Geyer et al. [45]. Univariate Cox proportional hazards regression models were fitted using the R package survival [46] and the parameter ties were set to “breslow”. Kaplan–Meier plots were constructed with the R package RTCGA [47]. The R package pROC were applied for plotting receiver operator characteristic (ROC) and calculating the Youden’s J statistic [48].

The partial least squares (PLS) regression and lasso logistic regression models were constructed using the R package caret [49]. The data were log_2_ transformed, centered and scaled. Only data with a definitive diagnosis and no missing data were included in the training and test data (*N* = 91). The data were split to ensure a balanced training set and 75% of the data was provided for training of the models and the remaining for testing the reported performance. The split of data were constructed to ensure a balanced training set. Highly correlated features and zero variance features were removed. The training model was optimized using 10-fold repeated cross validation and accuracy as performance metric.

### 2.7. Functional Analysis of Differentially Regulated Proteins

Functional analysis applying the hypergeometric function in R for dysregulated proteins against KEGG and gene ontology (GO) categories including cellular component (CC), biological process (BP), and molecular function (MF) are provided in Appendix A. GO and KEGG functional analyses were performed for all significantly regulated proteins (*p*-value < 0.05) obtained by the limma package in R, as described previously [43]. We compared MPE (*N* = 35) versus EBPE (*N* = 30) and MPE versus BPE (*N* = 57) samples. In addition, functional analysis was performed for all significantly up-regulated (*p*-value < 0.05) and significantly down-regulated proteins (*p*-value < 0.05).

The top 10 significant pathways were extracted from KEGG enrichment analysis results. In the case of GO, for each GO category, top 3 most significant terms of each GO category were extracted.

In Cytoscape, annotated enrichment maps of pathways for significantly regulated proteins (adjusted *p*-value < 0.05) were automatically generated. Significantly regulated proteins were identified by comparison of MPE samples (*N* = 35) versus BPE samples (N = 57) and MPE samples (*N* = 40) versus EBPE samples (*N* = 30). Briefly, significantly regulated proteins in R language programming (RCy3 package) were used to create a string network in Cytoscape with string protein query cutoff = 0.99 and a limitation query = 40. Then, functional enrichment was calculated in Cytoscape, and data were extracted in the format of “gmt.file”. After installation of the EnrichmentMap Pipeline Collection, the extracted “gmt.file” was used for EnrichmentMap generation. In the last step, autoAnnotate and subnetwork commands were used to create subnetworks based on the Markov clustering (MCL) algorithm.

### 2.8. Western Blotting Analysis

Aliquots of 20 µg of pleural effusion (PE) protein were separated by 10% SDS-PAGE. The proteins were transferred onto polyvinylidene fluoride (PVDF) membranes, and the membranes were blocked with phosphate-buffered saline with 0.1% Tween 20 (PBS-T) containing 5% skim milk, followed by incubation with primary antibody (1/10,000 dilution), overnight at 4 °C. Membranes were incubated with HRP-conjugated secondary antibody (goat anti-rabbit (1:10,000) 1/10,000 dilution) for 1 h at room temperature. Immunoreactivity was visualized with ChemiDoc Touch System (Bio-Rad, Hercules, CA, USA).

## 3. Results

### 3.1. Outline of Study

A total of 97 patients were enrolled prospectively in the study from 2019 to 2021. Malignant cancer status was updated in spring 2022. The probability of MPE in the cohort was 38%. Acellular PE was analyzed by LC-MS by at least two technical replicas resulting in a total of 214 LC-MS runs. Overall, 1689 protein isoforms were identified by LC-MS. The protein isoforms were collapsed into their coding genes resulting in 820 proteins identified in all PE groups. Two pairwise comparisons were performed: (1) all malignant versus all benign and (2) all malignant versus benign classified as exudate. Survival analysis was based on follow-up data obtained in spring 2022. Finally, pathways regulation and functional enrichment analysis were performed on the identified proteins and dysregulated proteins.

### 3.2. Demographic and Clinical Characteristics of Patients

A statistical summary of patients’ characteristics is outlined in Table 1 after follow-up data collection in spring 2022. In total, 97 patients including 35 MPE, 5 suspected but not confirmed malignant lung cancer and 57 BPE samples were enrolled. The malignant cases showed lower mean age compared to nonmalignant (Table 1). Gender was approximately equally balanced across the main clinical subgroups. Exudate PEs were strongly correlated with MPE. Cytology and clinical measurements of LDH and total protein concentration in PE were strongly correlated with malignant status. Exudate versus transudate classification is based on LDH and total protein PE measurements and the ratio to serum measurements. Although LDH and total protein concentration are statistically significant between MPE and BPE, there is a nonmalignant group of patients with similar total protein concentration as MPE (Figure 1), which are mainly EBPE. This is also evident for LDH (Figure A1). Consequently, we focus on the comparison between MPE and BPE for evaluation with other studies and in addition on MPE versus EBPE. The last pairwise comparison better reflects the clinical challenges. Regarding the tumor type, most lung cancer patients are diagnosed with adenocarcinoma histological type, with 62% confirmed cases out of all lung cancer cases.

### 3.3. Protein Identification

Across the 214 LC-MS runs, 1689 protein isoforms were identified. The protein isoforms were mapped to their corresponding genes resulting in 818 proteins. Figure 2a depicts the number of proteins identified factored on diagnostic status. The MPE group displays a high number of identified proteins despite considerably fewer samples in the MPE group compared to the BPE group. The MPE group also displays more unique proteins compared to the BPE group (Figure 2b). Extracellular region and extracellular vesicles (EVs) cellular component terms were found to have the highest functional enrichment when analyzing all identified proteins in both MPE and BPE (Figure 2c). Endocytic vesicle lumen, nucleosome, vesicle and focal adhesion were more enriched in MPE compared to BPE (Figure 2c).

### 3.4. Protein Dysregulation

Statistical analysis of quantitative proteomics data between MPE versus BPE (Appendix A) and MPE versus EBPE (Appendix A) revealed significant differences in 58/192 expressed proteins from MPE versus BPE and 65/218 DEPs from MPE versus EBPE (adjust *p*-value < 0.05/*p*-value < 0.05).

Volcano plots for the comparison between MPE versus BPE and MPE versus EBPE proteomes are provided in Figure 3a,b, respectively. Potential contaminant proteins from erythrocytes, platelets and coagulation cascade pathway obtained from Geyer et al. [45] were highlighted in red, orange and blue, respectively. Potential contaminant proteins displayed a minimal overlap with significantly dysregulated proteins (Figure 3a,b). LDHA and LDHB regulation was in agreement with clinical data and constituted protein malignant markers typically identified in lung fluids such as PE and bronchoalveolar lavage.

### 3.5. Survival Analysis Based on Protein Markers

Univariate Cox proportional hazards regression models were fitted to the observed survival, proteomics and clinical data for all lung patients in the cohort. The proportional hazard (PH) assumption was assessed using Schoenfeld residuals [50]. The Cox–Mantel log-rank tests for the top proteins (*p-*value < 0.05 and Cox–Mantel log-rank tests < 0.01) are listed in Table 2. None of the clinical measured variables correlated significantly to survival when applying the above thresholds. Figure 4 depicts the Kaplan–Meier for IGLV9_49 and proteasome activator subunit 1 (PSME1) which are the two most significant predictors for survival. Proteins such as PSME1 and HSP90AA1 have previously been associated with poor prognosis in cancer [51,52].

### 3.6. Comparative Classification Models

Different machine learning models were built based on clinical available parameters, proteomics data and a combination of clinical and proteomics data. Data from patients with a definitive diagnosis and no missing data (N = 91) were split into training (N = 52) and test data (N = 39) using 75% for training. Partial least squares (PLS) regression and lasso logistic regression were compared. PLS regression models demonstrated slightly higher accuracy than lasso logistic regression and were therefore explored in more detail (Figure 5). Receiver operating characteristics (ROC) analysis of PLS regression models demonstrated that models based on proteomics data increased performance in comparison to models based solely on clinical data. The highest area under the curve (AUC) was obtained by a combination of clinical and proteomics data (Figure 5a). PLS regression model trained on a combination of proteomics and clinical data maintained both clinical and proteomics features among the top 10 most important features (Figure 5b). Although, the clinical parameters total protein concentration and cytology examination were ranked as more important than the individual protein quantitative values. SLC3A2, GOT1 and BST1 obtained the highest importance in the PLS model (Figure 5b). The final PLS regression model using a combination of clinical data and PE protein quantitative data were evaluated by a confusion matrix (Figure 5c) and various performance measures calculated by caret R package (Figure 5d). The different performance measures are described in detail in [53].

### 3.7. Functional Enrichment for Significantly Regulated Proteins

To provide insight on the functional role of dysregulated proteins functional enrichment analysis using KEGG and gene ontology were performed. The analysis in Figure 6 was performed for the comparison between MPE and EBPE for protein significantly higher expressed in malignant. Gene ontology indicated enriched up-regulated proteins are mostly related to extracellular pathways (extracellular vesicular exosome, structural molecule activity and structural constituent of cytoskeleton, intermediate filament and keratin filament). Moreover, for KEGG pathway analysis revealed that metabolism-related pathways, especially glycolysis/gluconeogenesis, pyruvate metabolism and cysteine and methionine metabolism pathways were enriched among up-regulated proteins. In addition, the NOD-like receptor signaling pathway which is associated with infection and cancer were also identified.

In the comparison between MPE and all benign (BPE) proteome, the pathways related with metabolism such as glycolysis/gluconeogenesis, pyruvate and cysteine and methionine metabolism were significantly enriched among up-regulated proteins (Figure A2). In fact, these constitute the major biochemical pathways affected by cancer reprogramming, suggesting the relevance of proteins involved in metabolism in lung cancer PE physiology. Figure A3 summarizes the KEGG and GO analysis for down-regulated proteins. Extracellular vesicles and extracellular space are significantly enriched among down-regulated proteins. Furthermore, functions related to infection and inflammation were down-regulated for malignant samples (Figure A3).

The network enriched map of pathways of the significantly expressed proteins was investigated by the RCy3 R package and Cytoscape software (Figure A4). This provides a simplified overview of enriched pathways. In this approach, large networks were clustered into single nodes based on the Markov clustering (MCL) algorithm. MCL superiority in comparison with other methods of clustering has been demonstrated, particularly in showing the intra clusters’ edges [54,55,56]. 

Enriched map of pathways for comparison of the MPE samples versus EBPE identified 21, 18, 18, and 18 pathways in the membrane-bounded organelle, alpha-beta fibrinogen, complement receptor signaling, and intermediate filament rod domain clusters, respectively (Figure A4a).

Moreover, enriched map of pathways for comparison of the MPE samples versus BPE samples identified cancer signaling pathways, membrane-bounded organelle, lactate dehydrogenase domain, and plasma lipoprotein particles with 45, 25, 13, and 13 pathways as the four largest clusters, respectively (Figure A4b). This enriched map is in accordance with the functional enrichment result of both study groups (MPE versus EBPE and MPE versus BPE samples) and demonstrated the relevance of metabolism-related clusters such as lactate dehydrogenase domain and plasma lipoprotein particles.

## 4. Discussion

The survival parameters in this trial demonstrated an overall poor survival for malignant cases (Table 1). Furthermore, current clinical parameters used to identify malignant cases such as exudate versus transudate, total protein concentration, LDH and cytology correlated with histological confirmed malignant cases (Table 1). However, univariate assessment of these parameters is unable to clearly classify malignant cases (Figure 1c and Figure A1c). In the case of cytology, 46% of malignant cases are undetected (Table 1). The main challenge that persists is therefore to accurately classify exudate samples into malignant and nonmalignant.

Depletion of abundant proteins in body fluids such as albumin, immunoglobulins and related proteins prior to LC-MS was in the past considered as a requirement. We and others recently demonstrated that bronchoalveolar lavage [24] and plasma [22,57] fluids are possible to analyze without pretreatment prior to reduction, alkylation, trypsin digestion and LC-MS analysis. In this study, 1689 protein isoforms from 818 corresponding genes were identified from PE from non-depleted samples. The depletion step introduces sample processing bias, is time consuming and adds costs per sample. More proteins were identified from MPE samples compared to BPE (Figure 2a,b) even though the number of MPE samples is lower (35 compared to 57). This trend of higher proteome complexity in malignant samples is concordant with our previous findings for bronchoalveolar lavage [23,24]. Functional enrichment in cellular components based on all identified proteins in MPE and BPE suggested that PE is enriched in EVs (Figure 2c) as previously demonstrated for bronchoalveolar lavage [23,24]. This functional analysis also suggested nucleosome, focal adhesion and vesicle-related proteins as the main enriched proteins in malignant compared to nonmalignant cases for all identified proteins.

Pairwise comparison between MPE versus BPE and MPE versus EBPE identified multiple potential markers. Among them are well-established markers already applied in the clinic such as LDHA and LDHB. A subset was also identified as dysregulated in previous PE proteomics studies [32,34,58,59,60,61,62,63,64] which applied different sample preparation and MS methodologies (Figure 7). Various HSP90 family proteins were identified as potential biomarkers in this study by both performed pairwise comparisons (Figure 7). HSP90 proteins serve as poor prognosis markers in tissue of multiple cancers including lung cancer [51]. HSP90 proteins are involved in the stability of numerous proteins, including oncogenes such as kinase receptors such as EGFR [65]. Comparing proteins identified only in MPE with proteins significantly up-regulated MPE after correcting *p*-values for multiple testing again identified HSP90AA4P (Figure A5). Most proteins identified only in MPE are only identified in few samples. These proteins may result from stochastic sampling or from the expected high heterogeneity in malignant cases.

Structural molecule activity and intermediate filament terms were among the top up-regulated GO terms in the MPE versus EBPE proteome functional enrichment analysis. Collagen and actin fiber assembly proteins such as Biglycan (BGN), Elongation factor 1-gamma (EEF1G), Ras-related C3 botulinum toxin substrate 1 (RAC1), Myosin-2 (MYH2), Lamin type A (LMNA), and Catenin beta-1 (CTNNB) were among the top regulated. The role of BGN proteoglycan has been evaluated in lung tumor tissue microenvironment (TME) as a metastasis factor, recommending inhibition of stromal BGN as a tumor vascular normalization element [66]. Additionally, overexpression of EEF1G has been reported previously in lung cancer, and it showed a correlation with poor prognosis in patients [67]. Among the dysregulated genes in three independent cohort studies in the Oncomine database [68], EEF1G was detected as the poor prognosis protein in lung cancer. Additionally, RAC1, a family member of Rho GTPases, is capable of inducing epithelial-to-mesenchymal transition (EMT) and has a role in cell migration and metastasis through the activation of the PI3K/AKT signaling pathway [68]. LMNA is a scaffolding protein that contributes to the regulation of the cell cycle in lung cancer tumor cells [69]. AFM is a secreted glycoprotein and a member of the albumin superfamily of proteins that have been reported before in plasma, serum, cerebrospinal fluid, and follicular fluid [70], and it has been reported in various cancers such as ovarian and breast cancers [71,72]. This protein was among the up-regulated proteins comparing the MPE and BPE proteomes. As it was shown by the preliminary result in Figure A6, the Western blot experiments showed consistency with the proteomics results.

Furthermore, Univariate Cox proportional hazards regression model also identified various HSP90 proteins as associated with survival (Table 2). Extracellular HSP can promote cancer progression in breast cancer cell models [73]. The over-expression of HSPs has been observed in epithelial and hematological types of cancers, such as prostate, cervical, ovarian, renal, brain, lung, colorectal, hepatocellular, breast carcinomas, and myeloid leukemia [74]. Notably, immune-related proteins such as IGLV9-49 and PSME1 showed the highest association with survival (Figure 4), which pinpoints the importance of immune cells secretome and proteome for the outcome of patient treatment, particularly in the case of immunotherapy regimens and checkpoint inhibitor targeting drugs. The proteasome activator subunit (PSME) gene family has been shown to correlate with prognosis in gastric cancer [75]. To the best of our knowledge this study is the first to present a medium size PE proteomics cohort accessing protein markers association to survival. In total, 34 proteins were found significantly associated with survival (Table 2). The top protein survival markers were compared with tumor mRNA survival markers from lung adenocarcinoma (TCGA, PanCancer Atlas) [76] using cBioPortal [77]. Vitronectin (VTN) displays a highly significant correlation between mRNA expression and survival free progression (Figure A7). PSME1, HSP90AA1 and SERPINA3 display a similar trend as the protein markers from PE.

PLS regression models demonstrated that models based on quantitative proteomics demonstrate slightly higher AUC than models based on clinical data (Figure 5a). Furthermore, models based on proteomics data together with clinical data improved the AUC even further (Figure 5a). The selected markers in the model are previously associated cancer. For example, SLC3A2 and Ki67 are significantly correlated in NSCLC [50] and associated with poor prognosis in breast cancer [78]. GOT1 is involved in coordinating the glycolytic and the oxidative phosphorylation pathways in KRAS-mutated cancer cells [79]. BST1 is involved in immune suppression [80]. The current model based on clinical and proteomics data mainly suffer from imperfect precision. However, optimization of the quantitation of the protein markers may resolve this issue in the future. For example, by developing stable isotope-labeled references for absolute quantitation using for example multiple reaction monitoring. Currently the model based on combined proteomics and clinical data appears as a useful model to identify patients for frequent follow-up by for example image technologies.

Functional enrichment analysis provided insight into the dysregulated proteins in PE such as extracellular vesicles proteins, metabolism pathways and NOD-like receptor signaling pathway (Figure 6, Figure A2 and Figure A3, Appendix A). Extracellular vesicles proteins and metabolism pathways were also identified in our previous bronchoalveolar lavage proteome studies [23,24]. Additionally, metabolism proteome signatures with prognostic impact were previously identified in lung cancer tissue samples [81].

This study presents several limitations. Although the elimination of the depletion of abundant proteins reduced variance, it penalizes the total number of identified and quantified proteins. Given that EVs according to the functional enrichment analysis presented in this study are also abundant in PE, as was the case for bronchoalveolar lavage, a future study targeting EVs is planned. EVs from PE were isolated in our laboratory and we observe similar levels of EVs as in bronchoalveolar lavage (manuscript in preparation). In this study label-free quantitation with no reference proteins were applied for quantitation. Introducing isotope labeled standard peptides for target peptides of interest, may improve the accuracy of the quantitation and subsequent classification based on protein quantitative values. Moreover, we are currently developing immune-based methods for the quantitation of a panel of markers that can improve the AUC with a highest sensitivity and specificity compared to single marker (Figure A5). Another limitation of this study is the lack of image data on lung nodules to include in the machine learning models. Our future studies will address the EV proteome and target current identified markers with stable isotope label reference proteins analyzed by multiple reaction monitoring.

## 5. Conclusions

PE proteome is rich in highly significant survival markers. It appears based on comparisons with previous plasma-based studies [82] that PE proteome has slightly higher potential than plasma-based proteomics for classifying malignant cancers cases. Furthermore, PE proteome contains information that improves classification in comparison to only using clinical parameters and traditional laboratory measurements. Moreover, the proteins identified provide cues for establishing new treatment regimens targeted to patients with distinct molecular features based on the personalized patient protein signature.

## Figures and Tables

**Figure 1 cancers-14-04366-f001:**
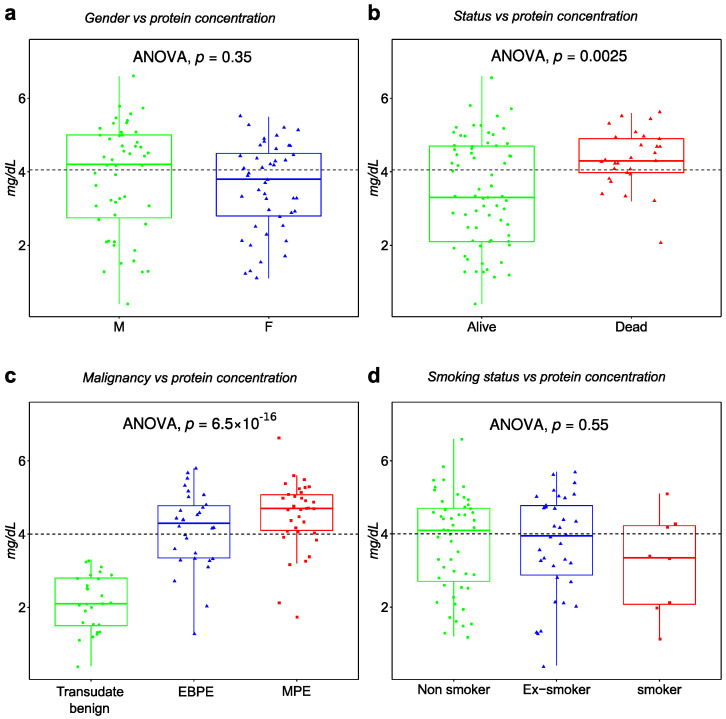
Boxplot of total protein concentration as function (**a**) gender, (**b**) survival status, (**c**) malignant status and (**d**) smoking status.

**Figure 2 cancers-14-04366-f002:**
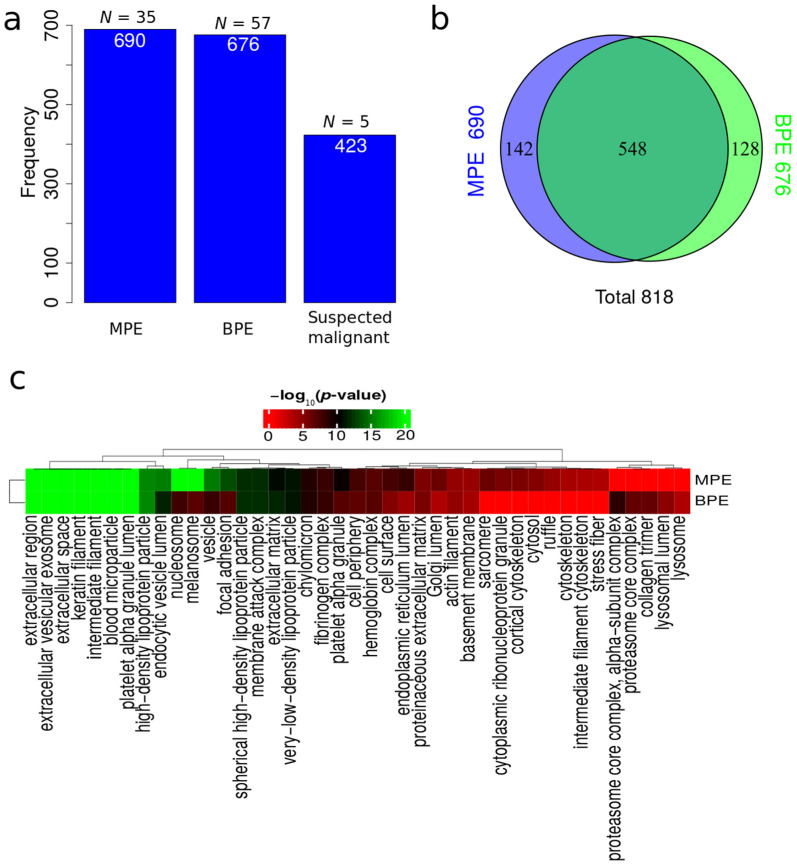
Summary analysis of identified proteins. (**a**) Total number of proteins identified in MPE, BPE and suspected malignant. (**b**) Venn diagram of identified proteins in MPE and BPE. (**c**) Functional enrichment analysis of cellular components based on all identified proteins in MPE and BPE.

**Figure 3 cancers-14-04366-f003:**
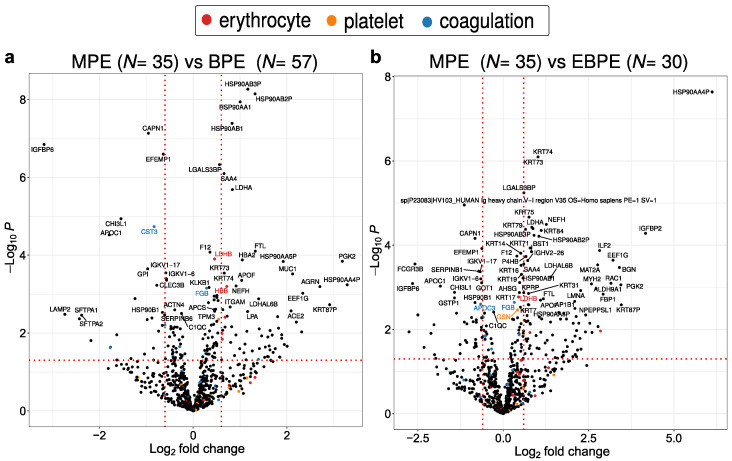
Volcano plot depicting dysregulated proteins when comparing (**a**) MPE versus BPE and (**b**) MPE versus EBPE. The *p*-value threshold of 0.05 is indicated by a horizontal red dotted line. Red vertical line depicts 1.5 fold up or down-regulation. MPE indicates malignant, BPE all benign and EBPE benign exudate.

**Figure 4 cancers-14-04366-f004:**
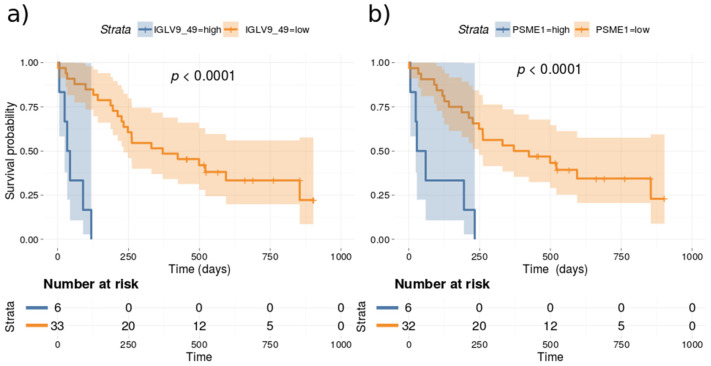
Kaplan–Meier based on dividing high and low expression of (**a**) IGLV9_49 and (**b**) PSME1.

**Figure 5 cancers-14-04366-f005:**
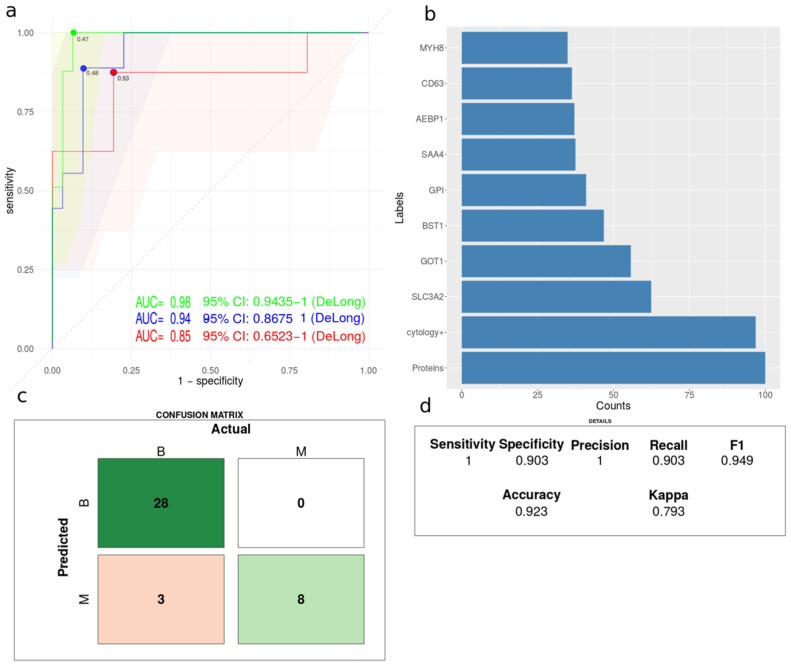
Performance of partial least squares regression analysis. (**a**) Receiver operating characteristics for three PLS regression models based on clinical data (red), label free quantitative proteomics data (blue) and clinical and label free quantitative proteomics data (green). The dot indicates the optimal cut of threshold based on Youden’s J statistic. (**b**) Histogram of the importance of the top 10 variables of PLS regression model trained on clinical and PE proteome data. The term “proteins” indicates total protein concentration. (**c**) Confusion matrix and (**d**) performance measures evaluated using the PLS regression model trained on clinical and PE proteome data and test data as input.

**Figure 6 cancers-14-04366-f006:**
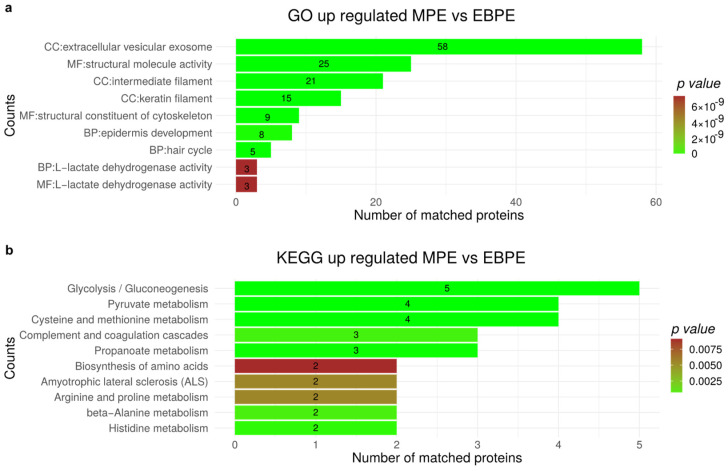
Functional analysis for significant up-regulated proteins by comparing MPE versus EBPE proteome. (**a**) Shows the significant GO analysis of up-regulated proteins (*p*-value < 0.05) between MPE and EBPE (CC, Cellular component; BP, biological process; MF, molecular function), (**b**) KEGG pathway enrichment analysis for up-regulated proteins by comparing MPE and EBPE.

**Figure 7 cancers-14-04366-f007:**
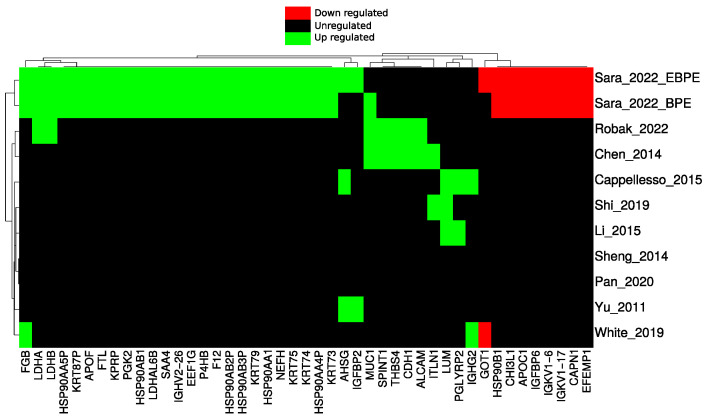
Heatmap of dysregulated proteins, separated into up, down and unregulated groups, comparing this study with previous PE studies.

**Table 1 cancers-14-04366-t001:** Summary of patients’ characteristics.

Group	Malignant (MPE)	Suspected Malignant	Non-Malignant (BPE)	*p*
Observations				
*N* = 97	35	5	57	
Age (Years)				
Mean (SD)	67 (15)	77 (9.5)	75 (15)	0.024
valid (missing)	35 (0)	5 (0)	57 (0)	
Gender				
F	51% (18)	60% (3)	42% (24)	0.56
M	49% (17)	40% (2)	58% (33)	
Pleural fluid				
Exudate	97% (34)	100% (5)	53% (30)	<0.001
Transudate	2.9% (1)		47% (27)	
Pleural LDH units				
Mean (SD)	372 (325)	237 (108)	266 (580)	<0.001
valid (missing)	34 (1)	5 (0)	57 (0)	
Pleural proteins mg/dL				
Mean (SD)	4.5 (0.97)	4.2 (0.3)	3.2 (1.4)	<0.001
valid (missing)	34 (1)	5 (0)	57 (0)	
Ethnicity				
Black	17% (6)	0% (0)	1.8% (1)	0.018
Caucasian	83% (29)	100% (5)	98% (56)	
Smoking status				
Ex-smoker	31% (11)	40% (2)	39% (22)	0.47
Non-smoker	57% (20)	20% (1)	53% (30)	
Smoker	11% (4)	20% (1)	5.3% (3)	
missing	0% (0)	20% (1)	3.5% (2)	
Cytology				
Negative	29% (10)	100% (5)	98% (56)	<0.001
Positive	54% (19)	0% (0)	0% (0)	
missing	17% (6)	0% (0)	1.8% (1)	
Status §				
Alive	29% (10)	20% (1)	100% (57)	<0.001
Dead	71% (25)	80% (4)	0% (0)	

§ Status 2.5 years post-PE collection.

**Table 2 cancers-14-04366-t002:** Cox–Mantel log-rank tests and *p*-value estimates from Cox proportional hazards regression models.

Protein	*p*	Log-Rank	Protein	*p*	Log-Rank
IGLV9_49	0.00014266	2.87 × 10^−5^	ACTC1	0.0048079	0.00023111
PSME1	0.00037219	1.68 × 10^−6^	ACTA2	0.00482817	0.00023881
HSP90AA1	0.0005116	0.00030706	SAA2	0.00527138	0.00336226
POTEKP	0.0005359	0.00012375	ACTA1	0.00547701	0.00029288
SERPINA3	0.00132541	0.00110944	LPA	0.006096	0.0044481
VTN	0.00148688	0.00096543	DSP	0.0064211	0.0015112
HSP90AB1	0.00165501	0.00114973	ITIH2	0.00652889	0.00572679
HSP90AB2P	0.00246227	0.00172132	ITIH4	0.00694899	0.00580105
HSP90AB3P	0.00248381	0.00175161	POTEE	0.00887962	0.00111277
SFTPD	0.00356403	0.00212968	POTEI	0.01041618	0.00164127
HPX	0.0036552	0.00398468	POTEF	0.01052107	0.00150148
ACTG1	0.00367013	0.00028455	SAA4	0.01084473	0.00922537
SAA2_SAA4	0.00397487	0.0029843	CPB2	0.01106905	0.0005958
HSP90AA5P	0.00400309	0.00297193	SAA1	0.0124629	0.00516249
ACTB	0.00401629	0.00031496	AKR1B10	0.01503984	0.00520114
H0YJW9	0.00436481	0.00344019	IGHV5_51	0.01558887	0.00675904
ACTG2	0.00448447	0.00020827	POTEJ	0.02431411	0.00718341

## Data Availability

The mass spectrometry proteomics data have been deposited to the ProteomeXchange Consortium via the PRIDE [83] partner repository with the dataset identifier PXD035884.

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
