# Peer review of "Assessment of a Large-Scale Unbiased Malignant Pleural Effusion Proteomics Study of a Real-Life Cohort"

_cancers, 2022, doi:10.3390/cancers14184366_

Round 1
Reviewer 1 Report
The study by Zahedi et al aimed to find protein markers of malignant pleural effusions potentially secreted by patients with lung cancer using mass-spectrometry based proteomics. The study found several high-confident proteins and therefore are of importance for researchers in the field of lung pathogenesis. However, I find that this study is better suited in proteomics-related journal.Also, I have several concerns especially with regards to number of replicates for the MS analysis, and lack of experimental validation of the identified proteins. I detailed the concerns below:
1. MS-based proteomics normally were done in triplicates seems there will be variation between MS runs. Please explain how they are 214 runs from the 97 samples. Also, the samples could be set aside for a few batches for validation purposes such as development of multiple reaction monitoring (MRM) or western blotting to confirm the MS findings.
2. Please change the word ‘novel’ as in your conclusion/introduction as each manuscript has its novelty. I found that these words were often used in authors’ other publications (e.g 10.1074/mcp.M113.034363 and 10.1038/srep42190)
3. The discussion can be more solid and not just repetition of results. The author should discuss the roles of identified proteins with previous reports, whether there are significant in lung cancer emergence; and whether there are newly discovered proteins. Some of the results also can be moved to discussion like in the section 3.5. Also can relate the role of the proteins with the functional enrichment figure (e.g. roles in extracellular vesicle)
4. Please give a better reference; not a Wikipedia page (line 400)
5. What are the protein expression levels of the identified protein sets in lung cancer patients? These can be compared using public datasets like TCGA cohort.
6. There are no experimental validations to support the MS identification of proteins. The author could run a simple western blot by selecting one or few target proteins since I assumed the samples are still available (collected in 50 ml? tube).
Author Response
Dear editor,
We would like to express our gratitude to the reviewers. Their comments help us improve our manuscript. We have addressed all the points to the best of our ability.
Kind regards,
Rune Matthiesen on behalf of the authors
Reviewer 1:
Comments and Suggestions for Authors
The study by Zahedi et al aimed to find protein markers of malignant pleural effusions potentially secreted by patients with lung cancer using mass-spectrometry based proteomics. The study found several high-confident proteins and therefore are of importance for researchers in the field of lung pathogenesis. However, I find that this study is better suited in proteomics-related journal.Also, I have several concerns especially with regards to number of replicates for the MS analysis, and lack of experimental validation of the identified proteins. I detailed the concerns below:
- MS-based proteomics normally were done in triplicates seems there will be variation between MS runs. Please explain how they are 214 runs from the 97 samples. Also, the samples could be set aside for a few batches for validation purposes such as development of multiple reaction monitoring (MRM) or western blotting to confirm the MS findings.
We thank for the comments. It is true that there is variance between LC-MS runs. However, typically only smaller mass spectrometry studies run the same biological replica in experimental triplicates. Clinical studies with many samples typically only run samples one time or as duplicates. The experimental design of our study is available in the PRIDE database but we have now also included the number of experimental replication we used in the method section of the manuscript to clarify this issue. We are currently working on optimizing the detection of the proposed biomarkers by antibody based methods and MRM. However, this is intended as patent application.
- Please change the word ‘novel’ as in your conclusion/introduction as each manuscript has its novelty. I found that these words were often used in authors’ other publications (e.g 10.1074/mcp.M113.034363 and 10.1038/srep42190)
We have rephrased our wording and removed novel/novelty from the manuscript. Please note that 10.1038/srep42190 was concerned with biomarkers in bronchoalveolar lavage whereas this manuscript analyzed pleural effusion.
- The discussion can be more solid and not just repetition of results. The author should discuss the roles of identified proteins with previous reports, whether there are significant in lung cancer emergence; and whether there are newly discovered proteins. Some of the results also can be moved to discussion like in the section 3.5. Also can relate the role of the proteins with the functional enrichment figure (e.g. roles in extracellular vesicle)
We have now extended the discussion and moved discussion paragraph of section 3.5 to discussion.
- Please give a better reference; not a Wikipedia page (line 400)
We have now included a review reference on machine learning performance measures.
- What are the protein expression levels of the identified protein sets in lung cancer patients? These can be compared using public datasets like TCGA cohort.
We have now compared our top protein survival markers from pleural effusion with survival analysis based on TCGA mRNA data from tumor tissue (Lung Adenocarcinoma (TCGA, PanCancer Atlas)). We find similar trends and for VTN high significance in terms of correlation to survival.
- There are no experimental validations to support the MS identification of proteins. The author could run a simple western blot by selecting one or few target proteins since I assumed the samples are still available (collected in 50 ml? tube).
We are working on optimizing the detection of the proposed biomarkers by alternative methods. We have now included preliminary data by Western blot that confirms our MS regulation.
Reviewer 2 Report
This study compares the proteomes of MPE and BPE with LC-MS based proteomic analysis and describes the biological functions enriched by differentially expressed molecules. The authors employed multiple models to fit the proteome data and clinical results, which identified several promising biomarkers for MPE. This study is well-written with comprehensive experimental design and analysis. It could be suitable to publish after minor revisions.
1. Please clarify the difference between BE and BPE. It seems BE samples are not included in 97 PE samples, so please describe the proteome data source of BE samples. Please keep consistent descriptions of MPE vs. BE or BPE in Fig6 and related paragraphs.
2. It seems different models identified different potential biomarkers (Table 2, Fig 5b), similar scenario for different publications (Fig7). The authors could propose the validation plans for these promising biomarkers in the discussion part.
3. It would be meaningful to investigate the MPE specifically identified/quantified proteins (Fig 2b) although these molecules are considered to be missing values in some situations.
Author Response
Dear editor,
We would like to express our gratitude to the reviewers. Their comments help us improve our manuscript. We have addressed all the points to the best of our ability.
Kind regards,
Rune Matthiesen on behalf of the authors
Reviewer 2:
Comments and Suggestions for Authors
This study compares the proteomes of MPE and BPE with LC-MS based proteomic analysis and describes the biological functions enriched by differentially expressed molecules. The authors employed multiple models to fit the proteome data and clinical results, which identified several promising biomarkers for MPE. This study is well-written with comprehensive experimental design and analysis. It could be suitable to publish after minor revisions.
- Please clarify the difference between BE and BPE. It seems BE samples are not included in 97 PE samples, so please describe the proteome data source of BE samples. Please keep consistent descriptions of MPE vs. BE or BPE in Fig6 and related paragraphs.
We agree that the abbreviation BE were not well chosen. BE was defined as exudate benign pleural effusion, whereas BPE is benign pleural effusion. That means that exudate benign pleural effusion is a subset of benign pleural effusion. We have changed the abbreviation for BE to EBPE to clarify this issue.
- It seems different models identified different potential biomarkers (Table 2, Fig 5b), similar scenario for different publications (Fig7). The authors could propose the validation plans for these promising biomarkers in the discussion part.
Thanks for the comment. We will include a discussion on how we intend to follow up on the current results. The different studies used different mass spectrometry methods and this influence which proteins will be detected and quantified (Figure 7). Concerning Table 2 and Figure 5B difference are expected. In table 2 we identified protein biomarkers correlating to survival. In Figure 5B we identified markers that correlate with the diagnosis of the patient. Interestingly, the markers for survival present stronger significance than the markers to final diagnosis of the patients.
- It would be meaningful to investigate the MPE specifically identified/quantified proteins (Fig 2b) although these molecules are considered to be missing values in some situations.
Thanks, we now include an analysis of the identified versus quantified proteins for MPE in the discussion.
Round 2
Reviewer 1 Report
Most of the concerns have been addressed. I do not have any further recommendation.